# Association between Accelerated Biological Aging, Diet, and Gut Microbiome

**DOI:** 10.3390/microorganisms12081719

**Published:** 2024-08-20

**Authors:** Shweta Sharma, Anna Prizment, Heather Nelson, Lin Zhang, Christopher Staley, Jenny N. Poynter, Gokul Seshadri, Aidan Ellison, Bharat Thyagarajan

**Affiliations:** 1Department of Laboratory Medicine and Pathology, University of Minnesota, Minneapolis, MN 55455, USA; sharm842@umn.edu (S.S.); prizm001@umn.edu (A.P.); sesha059@umn.edu (G.S.); ellis766@umn.edu (A.E.); 2Division of Epidemiology & Community Health, University of Minnesota, Minneapolis, MN 55455, USA; hhnelson@umn.edu; 3Division of Biostatistics & Health Data Science, University of Minnesota, Minneapolis, MN 55455, USA; zhan4800@umn.edu; 4Department of Surgery, University of Minnesota, Minneapolis, MN 55455, USA; cmstaley@umn.edu; 5Department of Pediatrics, Division of Epidemiology and Clinical Research, University of Minnesota, Minneapolis, MN 55455, USA; poynt006@umn.edu

**Keywords:** accelerated aging, gut microbiome, diet

## Abstract

Factors driving accelerated biological age (BA), an important predictor of chronic diseases, remain poorly understood. This study focuses on the impact of diet and gut microbiome on accelerated BA. Accelerated Klemera–Doubal biological age (KDM-BA) was estimated as the difference between KDM-BA and chronological age. We assessed the cross-sectional association between accelerated KDM-BA and diet/gut microbiome in 117 adult participants from the 10,000 Families Study. 16S rRNA sequencing was used to estimate the abundances of gut bacterial genera. Multivariable linear mixed models evaluated the associations between accelerated KDM-BA and diet/gut microbiome after adjusting for family relatedness, diet, age, sex, smoking status, alcohol intake, and BMI. One standard deviation (SD) increase in processed meat was associated with a 1.91-year increase in accelerated KDM-BA (*p* = 0.04), while one SD increase in fiber intake was associated with a 0.70-year decrease in accelerated KDM-BA (*p* = 0.01). Accelerated KDM-BA was positively associated with *Streptococcus* and negatively associated with *Subdoligranulum*, unclassified *Bacteroidetes,* and *Burkholderiales.* Adjustment for gut microbiome did not change the association between dietary fiber and accelerated KDM-BA, but the association with processed meat intake became nonsignificant. These cross-sectional associations between higher meat intake, lower fiber intake, and accelerated BA need validation in longitudinal studies.

## 1. Introduction

Chronic noncommunicable diseases, such as hypertension, coronary artery disease, and chronic respiratory diseases, become increasingly prevalent with accelerated biological aging, indicating that biological aging is a significant contributing factor [1,2]. While chronological age is a useful surrogate marker for the cumulative physiological dysregulations that occur with increasing age, it may not accurately reflect an individual’s functional status and susceptibility to aging-related diseases and disabilities [3]. To address this limitation, researchers have developed a metric of biological aging, such as the Klemera–Doubal biological age (KDM-BA) [4], which assesses multi-organ physiological dysregulation using standard clinically relevant biomarkers such as albumin, creatinine, and fasting glucose, among others, to provide a comprehensive measure of biological age and identify individuals at higher risk for age-related health issues. This biological aging measure has been associated with morbidity and mortality in diverse populations, including both healthy individuals and patients with heart disease, across a broad age range (20–85 years) [5,6,7,8]. The KDM-BA metric has been shown to be associated with other health indicators, such as multi-morbidity and activities of daily living, more strongly than other metrics of biological aging, such as DNA methylation-based epigenetic clocks and telomere length [9].

Diet plays a crucial role in influencing disease and accelerated biological aging. Diets rich in vegetables, fruits, nuts, cereals, fibers, fish, and unsaturated fats—containing antioxidants, vitamins, potassium, and omega-3—alongside reduced intake of red meat and ultra-processed foods, help prevent obesity, cardiovascular disease (CVD), and inflammation while promoting favorable glycemic, insulinemic, and lipidemic responses [9]. Studies have shown that adopting a Mediterranean diet, characterized by lower consumption of saturated animal fat and red meat and higher intake of fruits and vegetables, significantly impacts the body’s inflammatory state [10,11,12,13]. Additionally, lifestyle modifications such as engaging in physical and social activities and reducing smoking and alcohol consumption further contribute to overall health across various life stages, potentially delaying the onset of chronic diseases and promoting well-being [14].

The diet also impacts the gut microbiome, which may be a biological mediator in the relationship between diet and accelerated aging [15]. The gut microbiota is critical in regulating host biology and has been linked to the aging process and host dyshomeostasis in model animals [15]. Specific bacterial genera in the intestinal microbiome have been implicated in chronic diseases such as inflammatory bowel disease (IBD), type 2 diabetes (T2D), cardiovascular disease (CVD), and colorectal cancer [16,17,18]. Research from the United Kingdom reveals that gut microbiota differences are more closely correlated with biological age than with chronological age [19]. Another study links increasing biological age in community-dwelling adults to gastrointestinal dysbiosis [20]. Although there is consensus on the alteration of the gut microbiota composition with age and its association with aging [21,22], the influence of diet on the gut microbiome and its subsequent impact on biological aging remains underexplored [23,24].

This study aims to investigate the relationship between accelerated biological aging, as estimated by KDM-BA, diet, and the gut microbiome in participants of the 10,000 Families Study (10KFS). We hypothesize that accelerated biological aging will be positively associated with higher consumption of red/processed meat and negatively associated with higher intake of fiber, such as from fruits and vegetables [25]. Furthermore, we evaluate whether the gut microbiome mediates the relationship between diet and biological aging.

## 2. Materials and Methods

### 2.1. Study Population

The 10,000 Families Study (10KFS) is a new, family-based prospective cohort study in Minnesota that was established in 2017 with ongoing recruitment. A description of the cohort recruitment methods has been published previously [26]. Briefly, index participants were eligible for inclusion in the study if they were adults (≥18 years), lived in Minnesota, provided consent, and had at least one family member eligible to participate. Eligibility was assessed via a brief online eligibility screener. These ‘index participants’ were asked to invite family members to the study if eligible. Once at least one additional family member was enrolled in the study, the family became eligible for inclusion. The definition of “family” was determined by study participants. Participation in the study included completing an online questionnaire (self-completion for individuals 18 or older; parent completion for children <18 years), an invitation to attend a health visit, and providing a stool sample for microbiome analysis.

This study was approved by the Institutional Review Board at the University of Minnesota (IRB approval number: STUDY00000877). The current analysis includes the subset of 10KFS participants who enrolled in the study by the end of 2021, provided consent, questionnaire data, and a stool sample (n = 148). After further excluding people with missing diet information and accelerated KDM-BA (n = 31), 117 participants were included in the final analysis.

### 2.2. The 10KFS Health Visit

Biospecimens and other health measurements were collected during an in-person health visit conducted by trained study personnel. The study health visit included a collection of anthropometric measurements (height, weight, hip circumference, and waist circumference), blood pressure, pulmonary function, and biospecimens, including blood, urine, saliva, and samples as previously described [26]. In addition, participants were sent home with a stool collection Maxwell^®^ RSC Fecal Microbiome DNA Kit that was returned via mail.

Seated blood pressure and heart rate were measured using an Omron EVOLV (7000 series) upper arm blood pressure monitor. Total serum cholesterol was measured in serum using a cholesterol oxidase method (Roche Diagnostics, Indianapolis, IN 46250, USA) on a Roche Cobas 8000 Chemistry Analyzer (Roche Diagnostics Corporation). IgG antibodies to CMV were measured in EDTA plasma using the electrochemiluminescence immunoassay (on a Cobas 8000 Chemistry Analyzer, Roche Diagnostics Corporation). Spirometry for assessment of pulmonary function was performed with a FlowMIR Air Smart Spirometer (Pond Healthcare Innovation AB Lastmakargata 3, SE11144 Stockholm Sweden) and a disposable turbine with a cardboard mouthpiece (Medical International Research S.r.l. Via del Maggiolina 125 Rome Italy; USA MIR USA, Inc. 1900 Pewaukee Road, Suite D Waukesha, WI 53188, USA) and Air Smart Spirometer app (Pond Healthcare Innovation, Stockholm, Sweden).

### 2.3. Accelerated Klemera–Doubal Biological Age (KDM-BA)

The primary outcome for our analysis is accelerated biological age, measured as accelerated KDM biological age (KDM-BA), estimated as a continuous variable. We first calculated the biological age (KDM-BA) using chronological age and nine clinical biomarkers [4]: albumin, creatinine, fasting glucose, blood urea nitrogen (BUN), total cholesterol, systolic blood pressure, C-reactive protein, antibodies to cytomegalovirus (CMV) infection, alkaline phosphatase, and peak expiratory flow [4,27,28]. Subsequently, we determined “accelerated KDM-BA” by subtracting a person’s chronological age from their KDM-BA. If the KDM-BA is higher than their chronological age, it indicates that their biological age is accelerated, meaning their body shows signs of aging faster than expected for their chronological age. Conversely, if the KDM-BA is lower, it suggests that their biological age is decelerated, meaning their body is aging more slowly than expected for their chronological age.

### 2.4. Diet Assessment

Diet was assessed using the National Health and Nutrition Examination Survey (NHANES) diet screening questionnaire, which collected detailed information about food and nutrient intake [29].The 26-item Dietary Screener Questionnaire (DSQ) asks about the frequency of consumption of selected foods and drinks in the past month. The DSQ captures intakes of fruits and vegetables, dairy/calcium, added sugars, whole grains/fiber, red meat, and processed meat.

We used a SAS program provided by the NHANES [29] to calculate the predicted intake of fiber (gm) per day, calcium (mg) per day, whole grains (ounce equivalent) per day, total added sugar (tsp equivalent) per day, dairy (cup equivalent) per day, vegetables including legumes and French fries (cup equivalent) per day, vegetables including legumes and excluding French fries (cup equivalent) per day, and fruits (cup equivalent) per day. Meat variables were assessed using a self-reported questionnaire where participants reported the frequency of eating individual items in the following categories; “1-time last month”, “2–3 times last month”, “1 time per week”, “2 times per week”, “3–4 times per week”, “5–6 times per week”, “2–3 times per day”/“4–5 times per day”, “1 time per day” and “never”. For this analysis, the diet variables were converted to continuous variables ranging from 0 (never) to 7 (4–5 times/day).

### 2.5. Gut Microbiome

According to manufacturer recommendations, genomic DNA was extracted from human fecal slurry via Maxwell^®^ RSC Fecal Microbiome DNA Kit (Promega, Fitchburg, WI, USA). Previously published primers (Appendix A) targeting the V4 region of the 16s rRNA gene were acquired from IDT (Integrated DNA Technologies, Coralville, IA, USA) [30].

The amount of 2 µL of fecal microbial genomic DNA was amplified in 25 µL reactions using 3 µL of nuclease-free water, 12.5 µL of Platinum Hot Start PCR 2X Master Mix (ThermoFisher Scientific, Waltham, MA, USA), 5 µL of Platinum GC Enhancer (ThermoFisher Scientific, Waltham, MA, USA), and 1.25 µL of 100 µM primer for both the forward and reverse primers. DNA amplification was performed on a PCR thermocycler (Applied Biosystems™ MiniAmp™ Thermal Cycler, Waltham, MA, USA, Catalog number: A37834) with the following protocol: first denaturation at 94 °C for 2 min, followed by 35 cycles of 94 °C for 45 s, 50 °C for 60 s, and 72 °C for 60 s. The PCR product was purified via QIAquick PCR purification Kit (Qiagen, Hilden, Germany) according to manufacturer recommendations, and 4 nM libraries were pooled, denatured with 0.2 N NaOH, and diluted to 8 pM following the MiSeq v2 sequencing protocol (Illumina, CA, USA). Dual-indexed paired-end sequencing was performed using a MiSeq Reagent Kit v2 (500-cycles) cartridge on the Illumina MiSeq platform (Illumina, CA, USA).

After demulitiplexing, sequence data were processed using mothur ver. 1.41.1 [31], using a modified version of a previously published pipeline for V4 data [32]. Sequence data were quality-trimmed and aligned against the SILVA database (ver. 138) [32] for further processing. Chimeras were identified and removed using UCHIME ver. 4.2.40 [33]. Amplicon sequence variants were binned at 99% similarity using the furthest-neighbor algorithm. Taxonomy was annotated using the Ribosomal Database Project database (ver. 18) [34].

Three hundred and ninety-one genera were identified from 10KFS biological fecal samples collected from 148 participants. For this analysis, we only considered genera with a relative abundance of at least 1% in at least 80% of the samples, resulting in 266 genera. Finally, we included only 117 participants with complete information on KDM-BA and microbial genera.

### 2.6. Potential Confounders

Self-reported age, race/ethnicity, sex (male, female), smoking status, and alcohol intake were self-reported using a questionnaire at baseline. Smoking status was analyzed as a binary variable (current/former vs. never). Current and former smokers were combined into a single category due to the small number of current smokers in this study. Alcohol was recorded and analyzed as a categorical variable (current vs. former vs. never drinkers). Body mass index was calculated based on height (meters) and weight (kg) measured during the health visit using the formula “Weight (in Kg)/Height [2] (in meters)”. Family relatedness was self-reported by study participants at baseline.

### 2.7. Statistical Analysis

Data analysis was performed using SAS version 9.4. Descriptive statistics included means and standard deviations for continuous variables (accelerated KDM-BA, chronological age, BMI) and percentages/frequencies for categorical variables (sex, smoking, alcohol use). We used a univariate linear regression model (Model 1) to analyze the unadjusted association between accelerated KDM-BA and individual dietary components. Next, we evaluated the association between accelerated KDM-BA and individual dietary components after adjusting for covariates such as chronological age, sex, BMI, smoking status, alcohol use, and a random intercept for family relatedness using mixed effect regression (Model 2). These covariates were chosen based on their known associations with biological aging and dietary habits, supported by the existing literature [27,28]. Finally, we created a single model (Model 3) in which all dietary components significantly associated with accelerated KDM-BA in Model 2 (i.e., fiber intake, red meat intake, and processed meat intake) were further adjusted for bacterial genera associated with accelerated KDM-BA. This allowed us to evaluate whether adjustment for microbial genera associated with accelerated KDM-BA changed the association between dietary variables and accelerated KDM-BA.

To identify bacterial genera correlated with accelerated KDM-BA, we used Spearman correlation. Bacterial genera with significant Spearman correlation (r ≥ 0.2 or r ≤ −0.2 and *p* < 0.05) were further evaluated using a linear mixed effects regression model after adjustment for age, sex, BMI, alcohol intake, and smoking status, and using family relatedness as the random intercept. This helped us find bacterial genera significantly associated with accelerated KDM-BA. A similar method was used to identify the association between bacterial genera and dietary variables associated with accelerated KDM-BA. Given the large number of tests conducted, we applied the false discovery rate (FDR) correction to balance the discovery of true positives while controlling the expected proportion of false positives [35]. Bacterial genera with FDR-corrected *p*-values < 0.05 were considered significantly associated with accelerated KDM-BA.

Additionally, we estimated α diversity at the OTU (operational taxonomic unit) level using three indices: Shannon, Chao1, and Simpson’s. The Shannon index accounts for both OTU richness and evenness, Chao1 estimates OTU richness, including rare OTUs [36], and Simpson’s index assesses diversity with a focus on the dominance of common OTUs [37]. For β diversity, Bray–Curtis dissimilarities [37] were calculated in mothur and visualized by ordination with principal coordinate analysis. Analysis of similarity (ANOSIM) [38] was used to evaluate significant differences in β diversity. For γ diversity, we used the R statistical software’s vegan package, which measures the overall biodiversity of the gut microbiome across all participants.

## 3. Results

### 3.1. General Characteristics

The demographic characteristics of the participants included in this analysis are described in Table 1. This study included 117 individuals, with 78 women (53%) and a mean age of 54.34 +/− 17.49 years (age range: 19–91 years). Sixty-eight individuals (57%) showed decelerated KDM-BA (i.e., KDM-BA lower than chronological age). A vast majority of the study participants were white (94.87%) and nonsmokers (77.31%) (Table 1). There were 117 individuals in the study from 21 families; every individual was an adult. Each family is identified by a unique family identifier (FID).

### 3.2. Association between Accelerated KDM-BA and Diet

Univariate analysis showed that processed meat, red meat, calcium (mg/day), and dairy (cup equivalent/day) intake were positively associated with accelerated KDM-BA (Table 2). There was a 2.58-year increase in accelerated KDM-BA with one standard deviation increase in processed meat consumption (*p* < 0.001) and a 1.92-year increase in accelerated KDM-BA with one standard deviation increase in red meat consumption (*p* = 0.01) (Table 2, Model 1). There was a 0.01-year increase in accelerated KDM-BA with one standard deviation increase in calcium intake (*p* = 0.04) and a 1.95-year increase in KDM-BA with one standard deviation increase in dairy intake (*p* = 0.04) (Table 2, Model 1). However, after adjustment for family relatedness, sex, chronological age, BMI, alcohol intake, and smoking status, only processed meat intake (2.21 years increase in KDM-BA; *p* < 0.001) and red meat intake (1.62 years increase in KDM-BA; *p* = 0.03) remained significantly associated with accelerated KDM-BA (Table 2, Model 2). Additionally, fiber intake (gm/day) showed a significant inverse association with accelerated KDM-BA (0.56 years decrease, *p* = 0.04) (Table 2, Model 2). Including all significant diet variables from Model 2 (processed meat, red meat, and dietary fiber) in the same model after adjusting for covariates attenuated the association between processed meat and accelerated KDM-BA (1.91 years of accelerated KDM-BA; *p* = 0.04). Fiber intake remained inversely significantly associated with accelerated KDM-BA (a decrease of 0.70 years in accelerated KDM-BA; *p* = 0.01), while red meat was no longer associated with accelerated KDM-BA after adjustment for fiber and processed meat intake (*p* = 0.51). The *p*-values reported here are adjusted for multiple comparisons; the false discovery rate (FDR) correction was applied in the analysis to balance the discovery of true positives while controlling the expected proportion of false positives.

### 3.3. Association between Accelerated KDM-BA and Microbiome

Individual univariate linear regression analyses of the gut microbiome identified genera significantly correlated with accelerated KDM-BA. *Streptococcus* was positively associated with accelerated KDM-BA, while *Faecalitalea*, *Subdoligranulum*, and unclassified members of the *Bacteroidetes* (phylum) and *Burkholderiales* (order) were negatively associated with accelerated KDM-BA (FDR *p*-value < 0.05), suggesting an inverse relationship with age acceleration. Further adjustment for family relatedness, sex, chronological age, BMI, alcohol consumption, and smoking status did not change these associations (Model 2) except for *Faecalitalea*, which was no longer negatively associated with accelerated KDM-BA after multivariate FDR adjustment (Figure 1).

We found no association between α diversity indices (Shannon/Chao) and accelerated KDM-BA (Shannon: r = 0.15, *p* = 0.12; Chao1: r = 0.08, *p* = 0.41). Additionally, β diversity (differences in microbial community composition) did not differ significantly between individuals with accelerated KDM-BA and those with KDM-BA younger than their chronological age (ANOSIM R = 0.006, *p* = 0.296). Moreover, the γ diversity of 4.27 suggests that the gut microbiome is relatively diverse across all participants. This could imply that despite individual variations, the collective microbiome has a rich and varied composition.

### 3.4. Diet, Microbiome, and Accelerated KDM-BA

We did not identify any microbial genera significantly associated with meat or fiber intake. After adjusting for the microbial genera *Streptococcus*, *Subdoligranulum*, *Bacteroidetes* (phylum), and *Burkholderiales* (order)—which were found to be associated with accelerated KDM-BA but not directly linked to dietary intake of processed meat or fiber—the association between processed meat and accelerated KDM-BA was no longer statistically significant (1.49 years of accelerated KDM-BA; *p* = 0.07) (Table 2, Model 3). However, the association between fiber intake and accelerated KDM-BA remained unchanged after adjustment for the microbial genera (−0.65 years decrease in the accelerated KDM-BA; *p* = 0.01) (Table 2, Model 3).

## 4. Discussion

This is the first study to investigate whether the gut microbiome influences the association between diet (meat and fiber intake) and accelerated biological aging. Previous studies have examined the associations between diet (using dietary patterns and dietary indices) and microbiome [17,24,39,40], microbiome and biological aging (using frailty index, chronic kidney disease, and biological clocks) [21,23,41,42], and diet and biological aging [10,39], but have not examined all three components together in a single study. We estimated age acceleration using the difference between KDM-BA and chronological age. We found that increased dietary intake of processed and red meat was positively associated with accelerated KDM-BA, while higher fiber intake was inversely related to accelerated KDM-BA.

Our study investigated whether the gut microbiome mediated the association between diet and accelerated KDM-BA. We found that higher consumption of processed and red meat was associated with accelerated KDM-BA, while higher fiber intake was associated with a decrease in accelerated KDM-BA. These associations were only partly attenuated by adjustment for the gut microbiome, suggesting that the microbiome partly mediates the relationship between diet and accelerated aging. Specifically, the association between processed meat intake and accelerated KDM-BA was no longer statistically significant after adjusting for microbial genera associated with accelerated KDM-BA, indicating a mediating effect of the microbiome. In contrast, the inverse association between fiber intake and accelerated KDM-BA remained significant even after microbiome adjustment, suggesting that the gut microbiome does not mediate fiber’s beneficial effects on biological aging. None of the taxa were significantly associated with accelerated KDM-BA after adjusting for chronological age, BMI, sex, smoking status, and family relatedness. This finding suggests that the inverse association between fiber intake and accelerated KDM-BA is through mechanisms independent of the impact of fiber intake on the gut microbiome. Other possible mechanisms through which dietary fiber intake may negatively impact accelerated KDM-BA include anti-inflammatory effects or improved metabolic health [43,44,45].

Our findings on diet and biological aging (BA) are consistent with earlier studies, suggesting an association between diet and BA. A study conducted in the UK Biobank [34] showed an association between diet and accelerated biological aging (using phenotypical age (PA), another metric to measure biological age) in a model adjusted for chronological age; processed meat intake was associated with higher PA whereas plant-based foods had the opposite effect. That study found that consuming processed meat such as poultry, beef, lamb, and pork was associated with higher age acceleration (*p* < 0.01) [34]. On the other hand, a higher intake of cooked/raw vegetables, fresh/dried fruits, and cereal were associated with lower age acceleration (*p* < 0.01) [33] Another UK Biobank study found that consumption of grains (β = −0.252), vegetables (β = −0.044), and fruits (β = −0.179) was inversely associated with accelerated KDM-BA, while meat and protein alternatives (β = 0.091) had a positive association (all *p* < 0.001) with accelerated KDM-BA [46]. Our finding that fiber is inversely associated with accelerated KDM-BA is consistent with these previous studies, as the primary sources of dietary fiber are typically fruits, vegetables, legumes, and whole grains [47,48], which are all inversely associated with biological aging.

We found that the genus *Streptococcus* was positively associated with accelerated KDM-BA, suggesting that higher relative abundances of *Streptococcus* may be associated with accelerated biological age. Our findings about the *Streptococcus*–biological age association are consistent with a recently published study that utilized Mendelian randomization to show that *Streptococcus* was related to biological age acceleration [23]. Another study found an association between an increase in inflammatory marker C-reactive protein (CRP) and markers of metabolic dysregulation (e.g., glucose, HDL, Hemoglobin A1c) and an increase in *Streptococcaceae/Streptococcus infantis* [41], consistent with the association found in our study. The study that used a Mendelian randomization approach also reported that *Eubacterium rectale* (β = 0.20, *p* = 0.0190) and *Sellimonas* (β = 0.06, *p* = 0.019) were positively associated with accelerated KDM-BA, while *Lachnospira* (β = −0.18, *p* = 0.01) was inversely related to biological age acceleration [23]. Though our study could not confirm these associations with accelerated KDM-BA, the direction of association for all the remaining genera was similar to what was reported in the previous study.

We also found that *Bacteroidetes_unclassified, Burkholderiales_unclassified*, and *Subdoligranulum* were negatively associated with accelerated KDM-BA, suggesting that lower abundances of these microorganisms may be associated with accelerated biological age. A previous study on people living with HIV showed that higher levels of *Subdoligranulum* were associated with slower biological aging [49], consistent with our results. Additionally, people who follow diets rich in vegetables and fiber, such as Mediterranean dietary patterns, tend to have significant enrichment of *Bifidobacterium* and *Bacteroidetes* (especially *Prevotella*), which correlates with a lower inflammatory state and better general health status [50], supporting the inverse association between Bacteroides and accelerated KDM-BA observed in this study.

Limitations of this study include the cross-sectional study design, which makes it difficult to determine the temporality of the associations observed. Additionally, the sample size was relatively small, with only 117 participants, which increases the likelihood of finding associations by chance. It is also important to note that our sample consisted mainly of individuals with European ancestry, which may limit the generalizability of our findings to other populations. Finally, as our study was observational, there is a possibility of residual confounding. Further research, including randomized controlled trials and longitudinal studies, is crucial to clarify the causal relationships between diet, specific microbial taxa, and biological aging. These efforts will significantly contribute to a deeper understanding of the intricate interplay between the gut microbiota and aging processes, potentially leading to interventions that promote health.

## 5. Conclusions

This study reveals that diet significantly impacts biological aging, with processed meat accelerating and fiber reducing KDM-BA. We also found that the gut microbiome attenuates the effect of processed meat on accelerated aging, while fiber’s protective effects appear independent of gut microbial influence. These results suggest a complex interaction between diet, gut microbiome, and biological aging that must be considered when developing dietary interventions and microbiome modulation to promote healthy aging.

## Figures and Tables

**Figure 1 microorganisms-12-01719-f001:**
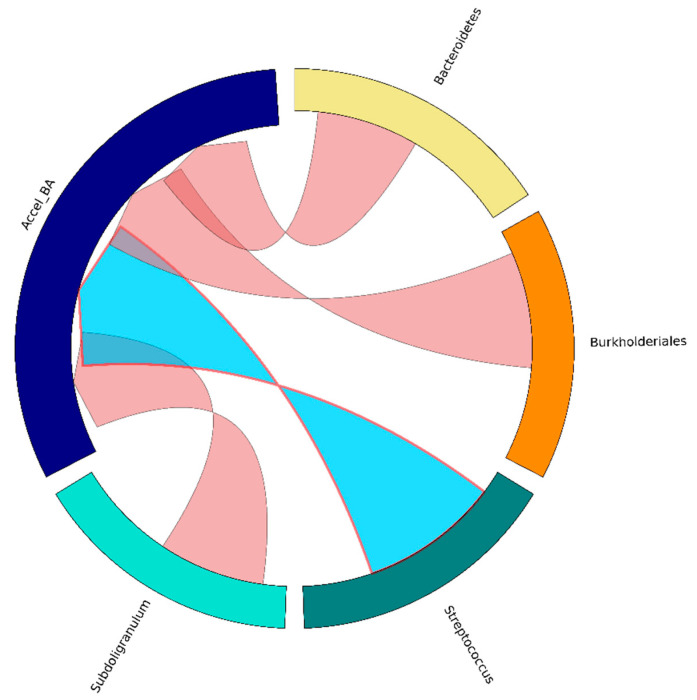
A Circos plot showing a linear association between accelerated KDM-BA (Accel_BA) and bacterial genera significantly associated with KDM-BA after adjustment for covariates (chronological age, sex, BMI, alcohol intake, and smoking status). Bacterial genera are positioned around the plot. Blue arrows symbolize positive linear relationships, indicating that an increase in accelerated KDM-BA corresponds to an increased relative abundance of the associated bacterial genus. Conversely, red arrows signify negative linear relationships, suggesting a decrease in the relative abundance of the bacterial genus with increasing accelerated KDM-BA. The directional arrow represents the strength and magnitude, and the broad arrow indicates stronger associations between the variables.

**Table 1 microorganisms-12-01719-t001:** Characteristics of study participants’ characteristics among 10KFS participants (n = 117) with accelerated KDM-BA (higher than chronological age (CA) and with decelerated KDM-BA (lower than CA).

Mean/SD and Percentage for Variables	Accelerated KDM-BA KDM-BA ≥ CA (n = 66)	Decelerated KDM-BA KDM-BA < CA (n = 51)	*p*-Value
Age	53.16/16.38	55.95/18.8	0.39
KDM-BA	47.87/15.58	61.95/20.20	<0.01 *
BMI	22.59/8.01	22.59/9.45	0.07
**Sex**			0.01 *
Women	51 (76.47%)	26 (50.98%)	
Men	15 (23.53%)	25 (49.02%)	
**Race**			0.63
White	62 (94.12%)	49 (96.08%)	
Other	4 (5.88%)	2 (3.92%)	
**Smoking status**			0.30
Never smokers	54 (82.35%)	38 (74.51%)	
Former/Current smokers	12 (17.65%)	13 (25.49%)	
**Alcohol intake**			0.93
Current drinkers	58 (87.88%)	44 (86.27%)	
Former drinkers	4 (6.06%)	4 (7.84%)	
Never drinkers	4 (6.06%)	3 (5.88%)	

SD—standard deviation. *—statistically significant *p*-value with alpha at 0.05.

**Table 2 microorganisms-12-01719-t002:** Association between diet and accelerated KDM-BA.

	Model 1		Model 2		Model 3	
Diet Variables	Beta Coefficient	*p*-Value	Beta Coefficient	*p*-Value	Beta Coefficient	*p*-Value
Processed meat	2.58	0.00 *	2.21	<0.001 *	0.21	0.07
Red meat	1.92	0.01 *	1.62	0.03 *	0.05	0.64
Fiber (gm) per day	0.06	0.79	−0.56	0.04 *	−0.33	0.01 *
Calcium (mg) per day	0.01	0.04 *	0.00	0.85	---	---
Whole grain (ounce equivalent) per day	1.50	0.38	0.23	0.90	---	---
Total added sugar (tsp equivalent) per day	0.08	0.66	−0.13	0.51	---	---
Dairy (cup equivalent) per day	1.95	0.04 *	1.04	0.33	---	---
Vegetables, including legumes and French fries (cup equivalent) per day	3.84	0.12	−1.22	0.67	---	---
Vegetables including legumes and excluding French fries (cup equivalent) per day	2.79	0.25	−1.36	0.61	---	---
Fruits (cup equivalent) per day	−1.14	0.50	−2.86	0.08	---	---
Added sugars from sugar-sweetened beverages per day	0.31	0.24	0.04	0.88	---	---
Fruits and vegetables, including legumes and French fries (cup equivalent) per day	0.24	0.83	−1.64	0.15	---	---

Model 1: Univariate analysis of individual dietary components with accelerated BA. Model 2: Linear mixed model evaluating the association between individual dietary components and accelerated BA after adjustment for family relatedness, BMI, gender, age, smoking status, and alcohol status. Model 3: Linear mixed model evaluating the association between significant dietary components (from Model 2 Table 2) and accelerated BA after adjustment for family relatedness, BMI, gender, age, smoking status, alcohol status, and significant gut bacterial genera that was found associated with accelerated KDM-BA. *—statistically significant *p*-value with alpha at 0.05.

## Data Availability

The data supporting this study’s findings are available upon request from the corresponding author, Bharat Thyagarajan.

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
