# Peer review of "Association between Accelerated Biological Aging, Diet, and Gut Microbiome"

_microorganisms, 2024, doi:10.3390/microorganisms12081719_

Round 1
Reviewer 1 Report
Comments and Suggestions for Authors
This great article examines the effect of the microbiome (in itself dependent on nutrients) on aging. The set-up is well done and the statistics are sound, while tables and figures are informative. I have some remarks:
- Table 2: without any correction, both Calcium and dairy worsen the biological age, which is quite baffling. Do you have an explanation for this finding?
- If the study population is supposedly representative, could gamma-diversity be calculated?
Author Response
COMMENT :Table 2: without any correction, both Calcium and dairy worsen the biological age, which is quite baffling. Do you have an explanation for this finding?
RESPONSE: As noted by the reviewer, the univariate analysis (Model 1) shows that dairy and calcium intake are positively associated with accelerated KDM-BA, while subsequent multivariate adjustment for other covariates (Model 2) shows no statistically significant association between dairy/calcium intake and accelerated KDM-BA. To address the question posed by the reviewer, we evaluated the confounding effect of each of the covariates included in Model 2 (i.e. BMI, gender, smoking status, and alcohol status) as shown in the tables below. These analyses showed that gender is an important confounder when evaluating the association between dairy and calcium intake and accelerated KDM-BA in that the beta coefficient for dairy and calcium intake are substantially attenuated after adjustment for gender.
Dairy intake
|
|
Beta coefficient |
|
Model 1 |
0.21 |
|
Model 1+ gender |
0.07 |
|
Model 1+BMI |
0.23 |
|
Model 1+smoking status |
0.20 |
|
Model 1+alcohol status |
0.20 |
Calcium intake
|
|
Beta coefficient |
|
Model 1 |
0.21 |
|
Model 1+ gender |
0.01 |
|
Model 1+BMI |
0.23 |
|
Model 1+smoking status |
0.20 |
|
Model 1+alcohol status |
0.20 |
Sex stratified analysis of dairy and calcium intake
|
|
Beta coefficient (Men) |
Beta coefficient (Women) |
|
Model 1 (Diary intake) |
0.12 |
0.37 |
|
Model 1 (Calcium intake) |
0.13 |
0.39 |
The sex stratified analysis suggests that the effect of diary and calcium intake on accelerated KDM-BA is stronger in women as compared to men. However, due to our limited sample size, we do not have sufficient power to evaluate the interaction between sex and diary/calcium intake on accelerated KDM-BA. This will need to be addressed in studies with larger sample sizes in the future.
COMMENT- If the study population is supposedly representative, could gamma-diversity be calculated?
RESPONSE: We have now calculated the gamma diversity as suggested by the reviewer and have included this metric in the revised manuscript.
“The gamma diversity is 4.27. Gamma diversity of 4.27 suggests that the gut microbiome is relatively diverse across all participants. This could imply that, despite individual variations, the collective microbiome has a rich and varied composition

Reviewer 2 Report
Comments and Suggestions for Authors
The manuscript " Association between accelerated biological aging, diet, and gut microbiome" presents results of an original study on searching relationship between diet, biological aging and gut microbiome. The main question addressed by the research is statistically supported association between gut microbiome features, diet pattern and intensity of biological aging. Generally, the study is scientifically sound and deserves to be published.
All results obtained by the authors are original and relevant for a field of human gut microbiomes. The paper addresses the specific gap in the data on presence of significant association between gut microbiome composition, diet pattern, and biological aging. The study adds new data on bacterial taxa in gut microbiome showing significant assotiation with acceleration or retardation of biological aging.
The conclusions are in a good agreement with the results and have a scientific soundness. All main questions posed and mentioned above were addressed with modern technique of high-throughput DNA sequencing, namely DNA metabarcoding, following with comprehensive bioinformatic and statistical analysis.
The authors have described comprehensively current state of research in biological aging, and have provided all necessary and appropriate references.
The text is well written, the figures are clear and comprehensive.
However, the manuscript needs some major and minor corrections.
1. The abstract does not reflect microbial findings described in the manuscript, namely the bacterial taxa that positively or negatively associated with biological aging.
2. Line 38. Term ‘association’ does not need Italic font.
3. Line 57. Choice of the Klemera Doubal Biological Age in contrast to others should be argued.
4. Lines 96-98. Hypothesis ‘that accelerated biological age will be negatively associated with increased consumption of red/processed meat and positively associated with higher intake of fruits and vegetables.’ means that MORE meat and LESS fruits and vegetables in diet relates to LOWER intensity of biological aging. But, it contradicts to the known observations showing unfavorable impact of red/processed meat as well as favorable influence of fruits and vegetables the human health and lifespan. So, such formulation of the hypothesis needs explanation or correction.
5. Lines 270-275. The contradiction between ‘we estimated α diversity at the OTU (Operational Taxonomic Unit)/ASV (Amplicon Sequence Variant) level…’ and ‘The Shannon index accounts for both genera richness and evenness, Chao1 estimates genera richness, including rare genera, and Simpson's index assesses diversity with a focus on the dominance of common genera.’
The authors should clearly indicate which taxonomic level they used for alpha-diversity estimation, OTU/ASV or genus.
6. Table 1. Sum of different drinkers (89+4+4) in the group with accelerated aging exceeds size of this group (66).
7. Line 317. Phrase ‘Fiber intake remained significantly associated with accelerated KDM-BA…’ contradicts phrase in Lines 311-312 ‘…fiber intake (gm/day) showed a significant inverse association with accelerated KDM-BA…’. It needs correction. I believe that ‘negative/inverse assotiation/correlation’ would be appropriate in this case.
8. Line 347. I believe that exchange of ‘was no longer associated’ to ‘was no longer negatively associated’ would be more exact and correct.
9. Two parts of the Discussion in lines 388-399 and 400-414 look like very similar and resulting in the same conclusions due to description of the same phenomena using slightly different phrases. For example, compare ‘Still, the association between fiber and accelerated KDM-BA remained unchanged after adjustment for microbial genera. This finding suggests that the association between fiber intake and accelerated KDM-BA is through mechanisms independent of the impact of fiber intake on the gut microbiome.’ and ‘…the inverse association between fiber intake and accelerated KDM-BA remained significant even after microbiome adjustment, suggesting that the gut microbiome does not mediate fiber's beneficial effects on biological aging.’. Thus, it is strongly recommended to combine these parts of the Discussion and correct them.
Author Response
COMMENT: The abstract does not reflect microbial findings described in the manuscript, namely the bacterial taxa that positively or negatively associated with biological aging.
RESPONSE: We have modified the abstract to include information about the bacterial taxa associated with biological aging. Specifically, we have included the following sentence in the abstract “Accelerated KDM-BA was positively associated with Streptococcus and negatively associated with Subdoligranulum, unclassified Bacteroidetes, and Burkholderiales”.
COMMENT Line 38. Term ‘association’ does not need Italic font.
RESPONSE: We have changed “association” to normal font in line 38.
COMMENT Line 57. Choice of the Klemera Doubal Biological Age in contrast to others should be argued.
RESPONSE: We chose the Klemera-Doubal Biological Age (KDM-BA) metric for our study due to its suitability for our hypothesis and the availability of necessary biomarkers. Unlike other methods such as epigenetic clocks and telomere length measurements, which require specific data like DNA methylation patterns and genomic analyses that we did not collect, KDM-BA utilizes clinically relevant biomarkers like albumin, creatinine, fasting glucose, and more, which were accessible in our dataset. This metric effectively captures physiological dysregulation across multiple organ systems, making it a comprehensive measure of biological age. Additionally, the KDM-BA metric has shown to be associated with other health indicators, such as multi-morbidity and activities of daily living, more strongly than other metrics of biological aging, such as DNA methylation-based epigenetic clocks and telomere length.
COMMENT Lines 96-98. Hypothesis ‘that accelerated biological age will be negatively associated with increased consumption of red/processed meat and positively associated with higher intake of fruits and vegetables.’ means that MORE meat and LESS fruits and vegetables in diet relates to LOWER intensity of biological aging. But, it contradicts to the known observations showing unfavorable impact of red/processed meat as well as favorable influence of fruits and vegetables the human health and lifespan. So, such formulation of the hypothesis needs explanation or correction.
RESPONSE: We have corrected lines 96-98 to state the following
“We hypothesize that accelerated biological aging will be positively associated with higher consumption of red/processed meat and negatively associated with higher intake of fiber, such as from fruits and vegetables”
COMMENT. Lines 270-275. The contradiction between ‘we estimated α diversity at the OTU (Operational Taxonomic Unit)/ASV (Amplicon Sequence Variant) level…’ and ‘The Shannon index accounts for both genera richness and evenness, Chao1 estimates genera richness, including rare genera, and Simpson's index assesses diversity with a focus on the dominance of common genera.’
The authors should clearly indicate which taxonomic level they used for alpha-diversity estimation, OTU/ASV or genus.
RESPONSE: We have clarified that the alpha diversity was calculated at the OTU level in the manuscript.
COMMENT Table 1. Sum of different drinkers (89+4+4) in the group with accelerated aging exceeds size of this group (66). The number are correct need to work on how to make it 119, it 148 now
RESPONSE: We apologize for this oversight and thank the reviewer for pointing out this error. We have now corrected table 1 to reflect the correct number of drinkers in the accelerated KDM-BA group. There are 58 current drinkers, 4 former drinkers and 4 never drinkers in our study.
COMMENT Line 317. Phrase ‘Fiber intake remained significantly associated with accelerated KDM-BA…’ contradicts phrase in Lines 311-312 ‘…fiber intake (gm/day) showed a significant inverse association with accelerated KDM-BA…’. It needs correction. I believe that ‘negative/inverse assotiation/correlation’ would be appropriate in this case.
RESPONSE: We have corrected the sentence to say the fiber intake was inversely associated with accelerated KDM-BA as suggested by the reviewer.
COMMENT Line 347. I believe that exchange of ‘was no longer associated’ to ‘was no longer negatively associated’ would be more exact and correct.
RESPONSE: We have made the change suggested by the reviewer.
COMMENT Two parts of the Discussion in lines 388-399 and 400-414 look like very similar and resulting in the same conclusions due to description of the same phenomena using slightly different phrases. For example, compare ‘Still, the association between fiber and accelerated KDM-BA remained unchanged after adjustment for microbial genera. This finding suggests that the association between fiber intake and accelerated KDM-BA is through mechanisms independent of the impact of fiber intake on the gut microbiome.’ and ‘…the inverse association between fiber intake and accelerated KDM-BA remained significant even after microbiome adjustment, suggesting that the gut microbiome does not mediate fiber's beneficial effects on biological aging.’. Thus, it is strongly recommended to combine these parts of the Discussion and correct them.
RESPONSE: We have combined these parts of the Discussion as suggested by the reviewer. We have now included the following sentences in the revised manuscript (lines 448-453).
“This finding suggests that the inverse association between fiber intake and accelerated KDM-BA is through mechanisms independent of the impact of fiber intake on the gut microbiome. Other possible mechanisms through which dietary fiber intake may negatively impact accelerated KDM-BA include anti-inflammatory effects or improved metabolic health”.

Round 2
Reviewer 2 Report
Comments and Suggestions for Authors
The manuscript " Association between accelerated biological aging, diet, and gut microbiome" presents results of an original study on searching relationship between diet, biological aging and gut microbiome. The main question addressed by the research is statistically supported association between gut microbiome features, diet pattern and intensity of biological aging. Generally, the study is scientifically sound and deserves to be published.
All results obtained by the authors are original and relevant for a field of human gut microbiomes. The paper addresses the specific gap in the data on presence of significant association between gut microbiome composition, diet pattern, and biological aging. The study adds new data on bacterial taxa in gut microbiome showing significant assotiation with acceleration or retardation of biological aging.
The conclusions are in a good agreement with the results and have a scientific soundness. All main questions posed and mentioned above were addressed with modern technique of high-throughput DNA sequencing, namely DNA metabarcoding, following with comprehensive bioinformatic and statistical analysis.
The authors have described comprehensively current state of research in biological aging, and have provided all necessary and appropriate references.
The text is well written, the figures are clear and comprehensive.
All major and minor corrections have been made by the authors.The revised manuscript has been improved significantly.